# Minced Beef Meat Paste Characteristics: Gel Properties, Water Distribution, and Microstructures Regulated by Medium Molecular Mass of γ-Poly-Glutamic Acid

**DOI:** 10.3390/foods13040510

**Published:** 2024-02-06

**Authors:** Mengmeng Qiao, Tao Zhang, Ming Miao

**Affiliations:** 1State Key Laboratory of Food Science and Resources, Jiangnan University, Wuxi 214122, China; 6210112062@stu.jiangnan.edu.cn (M.Q.); miaoming@jiangnan.edu.cn (M.M.); 2School of Food Science and Technology, Jiangnan University, Wuxi 214122, China

**Keywords:** γ-poly-glutamic acid, molecular weight, minced beef meat paste, gel property, water-holding capacity, microstructure

## Abstract

The influences of various m-γ-PGA (0.08–0.20%, *w*/*w*) concentrations on the properties of minced beef meat paste in terms of rheological properties, texture, moisture distribution, and microstructures were evaluated. The results indicated that m-γ-PGA enhanced the water-holding capacity, gel strength, texture, and whiteness of the minced beef meat paste. Based on the microstructural results, m-γ-PGA helped form a more organized and compact gel, thereby limiting the migration of water through the gel matrix. In contrast to the control group, the water-holding property, gel strength, and whiteness of minced meat paste gels with m-γ-PGA content of 0.12% increased from 75.89%, 584.51 g·cm, and 61.83 to 79.91%, 780.87 g·cm, and 62.54, respectively (*p* < 0.05), exhibiting the highest water-holding property and gel strength. Thus, m-γ-PGA exhibits great potential for minced meat paste products as a healthy gel water retainer and enhancer in low-fat meat products.

## 1. Introduction

Beef is an important high-quality food resource favored by consumers for its richness in proteins, micronutrients (iron, copper, zinc, vitamins A, B12, D, E, and other biologically active compounds), and low fat content [1,2]. Beef-based products include many products, such as dried, fried, smoked, roasted, marinated, fermented, sterilized, minced meat paste, etc. Minced meat products are popular among consumers because of their variety, tenderness, nutritional richness, and ease of consumption [3,4]. Nevertheless, minced beef products are prone to problems, such as water and oil loss during processing, resulting in a rough texture of the finished product, which affects product quality [5]. Starches, dietary fibers, non-meat proteins, and phenolic compounds have been identified for their ability to enhance the texture and water retention of meat batter products. For example, one study reported that carrageenan gum could effectively improve the water retention and textural properties of low-fat beef meatballs [6]. García-García and Totosaus [7] found that a mixture consisting of κ-carrageenan gum, potato starch, and acacia bean gum increased the yield obtained from cooking reduced-sodium and reduced-fat pork-beef sausage without affecting the sausage color. In addition, glutinous rice flour and deamidated and coupling-modified pea proteins performed well in beef patties in terms of improved water and oil retention and texture improvement [8,9]. However, introducing these substances may cause problems; for example, starch may bring excess calories, plant-derived proteins may trigger allergies, and excessive dietary fiber intake may affect the body’s mineral absorption. Therefore, identifying low-calorie, safe, and healthy strategies to respond to clean labeling is urgent [10].

γ-PGA is an edible, low-calorie, and non-toxic biomolecule with good biodegradability and biocompatibility. It, thus, has many application prospects in the current society that advocates for environmental protection and health [11,12]. γ-PGA is an unusual negatively charged allosteric peptide consisting of L-glutamic and D-glutamic acid monomers polymerized together by an γ-amide bond [13,14]. Originally found in Bacillus anthracis, γ-PGA is now mostly obtained by fermentation of Bacillus [15,16]. However, the primary emphasis of current studies regarding γ-PGA utilization has been directed toward pharmaceuticals and environmental management aspects [17,18], with less research on the γ-PGA utilization in food. Some researchers demonstrated the potential of γ-PGA as a dietary supplement to promote calcium absorption, a thickener, an antifreeze agent, a bitter taste reliever, and an oil-reducing agent in the food industry [19,20,21,22]. Furthermore, studies on γ-PGA potential as a texture improver for meat products have also been conducted. For example, γ-PGA is able to cross-link with the protein in chicken mince to a certain extent and has an excellent performance in improving gel characteristics and water retention of chicken mince [23]. Tao et al. [24] used γ-PGA as a cryoprotectant for grass carp mince and found that γ-PGA not only improves the protein’s ability to bind water molecules to inhibit the growth of ice crystals in the freezing process but also fills in the network structure of the protein and changes the cross-linking mode of the protein to a certain extent, which can better maintain the gel structure and sensory scores. Similarly, Hu et al. [25] found that in surimi protein gels, the lysine residues in myofibrillar protein molecules and the glutamic acid residues generated by γ-PGA thermal rupture can induce the formation of a more compact and homogeneous gel network structure through the cross-linking of non-disulfide bonds, which can improve the water retention of the gel.

The functional properties possessed by different molecular weight γ-PGAs also vary greatly, leading to different application scenarios. The same is true for the water retention properties of γ-PGAs. However, few articles state what the molecular weight is for γ-PGA to be used within the food system, and few studies reported on how to select the appropriate γ-PGA molecular weight for use in the food system. So, it is necessary to explain the selection of molecular weight of γ-PGA accordingly. In this paper, the γ-PGA with the strongest binding force to water molecules was quickly selected in the pure system by weighing method, and this molecular weight was chosen for the subsequent studies.

Previous studies have shown that γ-PGA was able to strengthen the protein network structure of minced fish and chicken by promoting cross-linking between proteins and performing padding, improving the water retention, texture, and antifreeze properties of the gels [23,24,25]. Therefore, we hypothesize that γ-PGA may have a positive effect on the gel properties of minced beef, but the relationship between the two is not clear. Hence, the goals of this research were to screen the γ-PGA molecular weight possessing excellent water retention effect and to explore the effects of the dosage of γ-PGA with this molecular weight on the water retention, gelatinization characteristics, and microstructure of minced beef meat paste. Meanwhile, our objective was to offer theoretical support for utilizing γ-PGA in enhancing the gelation quality of minced beef meat paste.

## 2. Materials and Methods

### 2.1. Materials and Ingredients

Fresh lean beef and salt were purchased from a local supermarket (Wuxi, Jiangsu, China). γ-PGA (1,303,672 Da) was provided by Shandong Jinyang Pharmaceutical Co., Ltd. (Zibo, Shandong, China). Food-grade sodium tripolyphosphate (STPP) was obtained from Hubei Xingfa Chemical Group Co., Ltd. (Xiangyang, Hubei, China). All the other chemicals were of analytical grade.

### 2.2. Preparation of Different Molecular Weights of γ-PGA

A solution of 80 g/L γ-PGA with 6 M HCl was adjusted to pH 2 and then heated at 80 °C with stirring for 1 and 4 h, respectively. After the reaction, the solution was rapidly cooled in an ice-water bath until it reached room temperature, and the reaction was terminated by adjusting the pH to 7 with 6 M NaOH. The γ-PGA hydrolysate solution was then dialyzed with distilled water for 48 h under 4 °C and was freeze-dried under vacuum conditions for 48 h to gain the powder sample of γ-PGA hydrolysate. The undegraded γ-PGA and degraded products with degradation times of 1 h and 4 h were referred to as high, medium, and low molecular weight γ-PGAs, which were labeled as h-γ-PGA, m-γ-PGA, and l-γ-PGA, respectively. Aqueous solutions of 1 mg/mL of the three products were made and then filtered through the 0.22 μm membranes for subsequent molecular weight determination.

### 2.3. Molecular Weight Analysis

The molecular weight (Mw, Mn) and molecular weight dispersion coefficient (PDI) of the degradation products were analyzed using Gel Permeation Chromatography (GPC) [26]. The test conditions were as follows: a TSK gel G3000SWXL column (7.8 mm × 300 mm); mobile phase 0.1 M Na_2_SO_4_–0.1 M Na_3_PO_4_ (pH 6.7); column temperature 30 °C; flow rate 0.5 mL/min; and a 2487 UV detector (220 nm).

### 2.4. Moisturizing Evaluation

Different water retention agents have different forces on water molecules and different abilities to retain water. A water retention agent with a strong force on water molecules will keep more water. The water-containing sample was placed in a constant temperature and humidity environment, and the change in the sample mass was weighed at regular intervals to calculate and compare the moisture retention of the sample. The slower the sample mass decreases, the stronger the binding ability of the sample to the water molecules and the better the water retention effect of the sample. This specific method is as follows: γ-PGA with three molecular weights was prepared as 2 mg/mL aqueous solutions. Approximately 3.0 g of the aqueous solution was weighed into a glass weighing flask that had been subjected to constant weight treatment. The glass weighing flasks containing the samples were placed open in a dryer at constant temperature and humidity (43 ± 2% relative humidity, 25 °C) and weighed at regular intervals [27]. Moisture retention was calculated using Equation (1):(1)$$Moisture\;retention\;(\%)=\frac{m_{t}}{m_{0}}\times 100\%$$
where *m_t_* means the mass of the sample that has been left for *t* h, and *m*_0_ is the initial sample mass (g).

### 2.5. Minced Meat Paste Preparation

Minced meat paste preparation followed the steps of Wang et al. [28] with some minor modifications. Briefly, any noticeable fat and connective tissue were eliminated and the meat was cut into small cubes. The meat cubes were chopped using a mixer (S30-LA598; Joyoung Co. Ltd., Hangzhou, Zhejiang, China) for approximately 3 min. After adding 0.2% (*w*/*w*) STPP, 1.5% (*w*/*w*) salt, and 1/3 ice water (15% total ice water, *w*/*w*), the minced meat paste was blended for 3 min and rested for 1 min. To the mixture, 2/3 ice water and the required amount of m-γ-PGA (0, 0.08, 0.12, 0.16, and 0.20%, *w*/*w*) were added and stirred for about 2 min. The temperature remained below 10 °C during the entire process.

In each group, the prepared minced beef paste was evenly distributed into 50 mL polypropylene centrifuge tubes (approximately 35× *g* of minced meat paste). The tubes were subsequently sealed and centrifuged at 2000 rpm for about 5 min at 4 °C (TGL-16M, Hunan Xiangyi Laboratory Instrument Development Co. Ltd., Changsha, Hunan, China) to remove residual air bubbles. The water bath (HH-2, Changzhou Ronghua Instrument Manufacturing Co. Ltd., Changzhou, Jiangsu, China) was used to heat the partially centrifuged minced meat paste for each treatment for 30 min to form a gel at 80 °C. The heated minced meat paste gel was then rapidly cooled for 10 min using an ice water bath and then stored at 4 °C. These samples were subsequently used for water-holding capacity (WHC) and cooking loss determinations, puncture tests, texture evaluation, moisture status and distribution assays and instrumental color and microstructure observations. The rest of the uncooked minced meat paste was refrigerated at 4 °C before dynamic rheology evaluation.

### 2.6. Rheological Properties

The rheological behavior of minced beef paste was tested using a rheometer (DHR-3, Waters Co., Ltd., New Castle, DE, USA), using the methodology described by Huang et al. [29], with minor adjustments. The uncooked samples were placed at the bottom of the platform. The parallel-plate probe utilized in this study had a diameter of 40 mm, maintained a distance of 1 mm from the specimen, and applied a strain of 0.5%. Dynamic frequency scan: A scan was performed at 25 °C ambient temperature, covering frequencies from 0.1 to 100.0 Hz. The damping factor (tan δ) and the storage modulus (G′) were gathered. Dynamic temperature scan: The temperature underwent a sweeping change, starting from 25 °C and reaching 80 °C at a ramp rate of 2 °C/min with 0.1 Hz oscillation frequency. Subsequently, the temperature was reduced to 25 °C at a rate of 5 °C/min. The damping coefficient (tan δ) and the storage modulus (G′) during the experiment were noted. The portion exposed to the air was sealed with silicone oil to prevent moisture loss.

### 2.7. Puncture Test of Gel

The method for determining gel strength was adopted from Yi et al. [30]. The prepared gels were allowed to equilibrate at ambient temperature for 2 h and sliced into cylinders (2.5 cm diameter and 1 cm height). A texture analyzer (TA-XT Plus, Stable Micro Systems Co., Ltd., Godalming, Surrey, UK) with a spherical probe (P/0.25 s) was used to measure the gel strength. The test parameters were as follows: test speed of 1 mm/s, sample penetration distance of 7 mm, and trigger force of 5 g. The gel strength is the product of the destructive force and destructive distance when puncturing the gel.

### 2.8. Texture Profile Analysis (TPA)

A texture analyzer (TA-XT Plus, Stable Microsystems Ltd., Godalming, Surrey, UK) was used to conduct TPA [31]. Samples were equilibrated under ambient temperature for 2 h and subsequently sliced into 1.5 × 1.5 × 1 cm (length × width × height) rectangles. The specific test conditions were as follows: P36R test probe; trigger force 5 g; compression twice with 50% compression rate; pre-test and test speeds of 2 mm/s; and post-test speed of 5 mm/s. TPA parameters, including cohesiveness, springiness, gumminess, hardness, and chewiness were recorded.

### 2.9. Cooking Loss and Water-Holding Capacity (WHC)

Cooking loss was measured according to Xing et al. [32]. The initial mass of the minced meat paste prior to cooking was denoted as *W*_1_, and the mass of the resulting cooked meat gel prepared according to step 2.5 was denoted as *W*_2_. Equation (2) was used to calculate the cooking loss:(2)$$Cooking\;loss\;(\%)=\frac{W_{1}-W_{2}}{W_{1}}\times 100\%$$

The WHC of the meat gel was estimated by centrifugation [33]. The cooked gel samples were sliced into sections of approximately 5 mm thickness and weighed precisely to record the mass as *M*_1_. The sliced samples were enveloped in filter paper, positioned within 50 mL centrifuge tubes, and centrifuged at 5500 rpm for 15 min at 4 °C (TGL-16M, Hunan Xiangyi Laboratory Instrument Development Co. Ltd., Changsha, Hunan, China). After centrifugation, the filter paper was removed, and the mass of the gel was measured again and recorded as *M*_2_. The WHC was calculated using Equation (3):(3)$$WHC\;(\%)=\frac{M_{2}}{M_{1}}\times 100\%$$

### 2.10. Distributions of Water Molecules

Low-Field Nuclear Magnetic Resonance (LF-NMR) was used to analyze the binding state and distribution of water molecules within the meat products. LF-NMR measurements were conducted using an NMR analyzer (MesoMR23-060V-I, Niumag Electric Co., Shanghai, China), and the test methodology was slightly modified, as described by Ding et al. [34]. The gel sample (approximately 2 g) was placed into an LF-NMR tube (25 mm diameter) and then into the instrument. The test temperature was 32 °C, and the frequency of proton resonance was 21 MHz. The transverse relaxation time (T_2_) of the gel samples was measured using the Carr–Purcell–Meiboom–Gill (CPMG) sequence. The parameters used for the CPMG sequence were as follows: SW = 100 KHz; TW = 4500 ms; TE = 0.3 ms; NECH = 18,000; NS = 2; and PRG = 1. The CPMG exponential decay curves were inverted and fitted using Multi Exp Inv Analysis software (version 1.0; Niumag Electric Corporation, Shanghai, China). T_2b1_, T_2b2_, T_21_, and T_22_ were the four lateral relaxation times, while P_2b1_, P_2b2_, P_21_, and P_22_ denoted their corresponding ratios of peak areas, from which the binding state and distribution of water molecules within the meat products could be obtained.

### 2.11. Color Evaluation

The thickness of the gel sample required for testing was 10 mm, and the diameter was 25 mm. Brightness (*L**), yellowness (*b**), and redness (*a**) were measured using a portable spectrophotometer (NS800; Shenzhen 3nh Technology Co., Ltd., Shenzhen, China). The test parameters were a viewing angle of 10°, an aperture of 8 mm, and a light source of D65 [35]. Whiteness (W) values were calculated using Equation (4):(4)$$Whiteness=100-\sqrt{{\left(100-L^{*}\right)}^{2}+a^{{*}^{2}}+b^{{*}^{2}}}\times 100\%$$

### 2.12. Scanning Electron Microscopy (SEM)

The scanning electron microscopy (SEM) method used was slightly modified from that of Zhao et al. [36]. Samples were cut into 3 × 3 × 3 cm squares and fixed in 2.5% glutaraldehyde (containing 0.1 M phosphate buffer) for 24 h. The fixed samples were subsequently submerged in a phosphate buffer solution (0.1 M, pH 7.2) for approximately 10 min and rinsed three times with the same phosphate buffer. Dehydration was performed using ethanol at different concentrations (30, 50, 70, 90, and 100%, *v*/*v*) for 20 min each, and 100% (*v*/*v*) ethanol was added three times. The samples were then submerged in acetone for deep dehydration and dried at 50 °C in an oven for 15–20 min. Gold was sputter-plated onto the samples and visualized using SEM (SU8100, Hitachi High-Technologies Corp., Tokyo, Japan). The accelerating voltage was set at 3 kV.

### 2.13. Statistical Analysis

All measurements were repeated at least thrice, and the results were presented as mean ± standard deviation (SD). Data were processed and plotted using SPSS statistical software (version 25.0; SPSS Inc., Chicago, IL, USA) and Origin 2021 Pro (Origin Lab, Northampton, MA, USA). Data were analyzed using a one-way analysis of variance (ANOVA), and multiple comparisons were performed using Duncan’s Multiple Range Test (DMRT) with a significance level of 0.05 through SPSS 25.0 software. Furthermore, principal component analysis (PCA) and Pearson’s correlation analyses were also performed using the relevant application software in Origin 2021 Pro.

## 3. Results and Discussion

### 3.1. Preparation of Different Molecular Weights of γ-PGA

In order to evaluate the effects of γ-PGA molecular weight on the properties of minced meat paste, we prepared several molecular weight γ-PGA degradation products by a previously laboratory-optimized high-temperature acid degradation. The molecular weights of the degradation products decreased with the degradation time, and the medium and the lowest molecular weights obtained from the preparation were 731,502 Da and 340,963 Da, respectively (Table 1). Weight average molecular weight (Mw) and number average molecular weight (Mn) are the two most important indexes for examining the molecular weight of polymers; the calculation of the former is mainly affected by the high molecular weight components, and the latter is primarily affected by the low molecular weight components, so the heavy average molecular weight is generally chosen as the result of the characterization of large molecules [37]. The polydispersity (PDI) reflects, to some extent, the difference between polymers with the same basic chemical structure but different polymerizations, and it is the ratio of Mw to Mn. The smaller the PDI, the more homogeneous the product. The polydispersity coefficients of the γ-PGA degradation products were small, indicating that the products have better homogeneity and higher application value.

### 3.2. Selection of Degradation Products of γ-PGA with Strong Moisturizing Property

The weight of the samples decreased with time, but the moisturization rate of m-γ-PGA was always higher than the other two groups at the same time (Figure 1). This may be because after hydrolysis, compared with the undegraded h-γ-PGA, m-γ-PGA exposed more hydrophilic groups, which were more likely to associate with neighboring water molecules to form hydrogen bonds, thus inhibiting water dissipation [38]. At the same time, the m-γ-PGA molecular chains can cross and entangle, forming a complex three-dimensional mesh structure to wrap the water molecules in it, further hindering the movement of water molecules [39]. Due to the lower molecular weight, l-γ-PGA contains fewer hydrophilic groups compared to m-γ-PGA, so the number of hydrogen bonds formed with water molecules is lower, resulting in a lower moisturization rate than m-γ-PGA. In summary, m-γ-PGA has a more excellent moisturizing effect. Therefore, we chose m-γ-PGA for the subsequent experimental study of water retention and gelation effects on minced meat paste gel products.

### 3.3. Rheological Properties during the Gelation Process

In the dynamic rheology test, the storage modulus (G′) value indicates the gel strength of the minced meat paste, with higher G′ values reflecting better gelation ability [40]. The intricate relationship between m-γ-PGA varying concentrations and minced beef paste G′ values at different frequencies are displayed in Figure 2a. The G′ values showed an increasing trend to different degrees as the oscillation frequency increased for all samples. The G′ values of all experimental groups increased more in the smaller oscillation frequency range (0.1–10 Hz). As the frequency increased, the protein molecules within the minced meat paste were entangled with each other and made a stable gel network, resulting in smaller changes in the G′ value. Moreover, at the same oscillation frequency, minced meat paste with m-γ-PGA demonstrated higher G′ values in comparison to the control group. Notably, the minced meat paste containing 0.12% m-γ-PGA displayed the highest G′ value.

The ratio of G′ to G″ defines the damping factor (tan δ). G″ represents the loss modulus, also referred to as the viscous modulus, indicating the sample’s viscosity properties. A larger tan δ indicates that the sample is more mobile and exhibits a viscous fluid behavior. On the contrary, the smaller tan δ, the weaker the fluidity and the higher the elasticity, and the sample behaves as a sol or gel solid [29]. Figure 2b illustrates a rise in the samples’ tan δ with an increasing oscillation frequency, but none of them exceeded 1, indicating that the G′ value consistently exceeded the G″ value, and the overall dominant performance was always the elastic property. The minced meat paste containing 0.12% m-γ-PGA exhibited the lowest tan δ in comparison to the control group, showing that the m-γ-PGA at this concentration could effectively enhance the gel system’s elasticity by improving the gel structure [41].

The heating of minced meat paste is an unstable dynamic rheological process accompanied by physical and chemical reactions, such as protein unraveling, denaturation, and coagulation. Detecting the changes in G′ over kinetic heating can be used to show the unfolding and aggregation of proteins, reflecting the changes in protein structure at different times [42]. The trend of G′ values was essentially the same for all samples throughout the heating process in Figure 2c. At a temperature range of 30–43 °C and when the samples contained m-γ-PGA at a concentration of 0.12%, G′ values decreased slowly in all groups, with a shift in the peak temperature from 48.72 to 46.56 °C in the control group. After G′ reached its maximum value, it began to decrease sharply with the temperature rising. In a temperature range of 51–56 °C, G′ reached the second peak for all the samples. G′ decreased because the protein began to denature, and myosin tails gradually unfolded as the temperature increased, causing a temporary collapse of the protein network structure. The disintegration of the protein mobility was enhanced, causing the G′ to decrease drastically [3]. Samples with m-γ-PGA added had significantly larger G′ minimums and lower transition temperatures than the control. As temperatures increased to >60 °C, G′ increased sharply. This is mainly due to the cross-linking of proteins by disulfide bonds and hydrophobic interactions to form orderly and irreversible gels [4,43]. When the gelation endpoint temperature was attained, it was noted that adding m-γ-PGA resulted in a higher G′ value for the minced meat paste than the control group. This suggests that m-γ-PGA positively influenced gel network formation. However, with the increase in m-γ-PGA amounts, G′ tended to increase and then decrease. These findings suggest that electrostatic interactions facilitated the binding of myosin’s positive charge and m-γ-PGA’s negative charge when the m-γ-PGA amount was relatively low, leading to a conformational change in myosin and causing a lower denaturation temperature of the myosin head further. When m-γ-PGA concentration was high, the excessive negative charge brought about led to excessive electrostatic repulsion between protein molecules, thus adversely affecting the gel [44].

### 3.4. Influence of the m-γ-PGA Content on Gel Strength

Gel strength is frequently used to assess the gel quality of ground beef slurries. The elasticity and hardness of the gel are related to breaking deformation and force, respectively [34]. In Figure 3, with the rise in m-γ-PGA amount, gel strength and fracture force of the meat gels tended to increase and then decrease. In contrast to the control group, the gels exhibited the highest fracture force and gel strength at a m-γ-PGA content of 0.12%, measuring 1231.43 g and 780.87 g·cm, but the gels’ fracture deformations were only slightly increased. The gel strength exhibited a similar pattern to the final G′ value observed during the previous rheological analysis after heating. The increase in gel strength could potentially be linked to the random breakage of the polypeptide chains after m-γ-PGA heating. This occurred during the gel formation by heating of minced beef paste and cross-linking between Glu residues exposed by degradation of m-γ-PGA and Lys residues in the proteins, leading to an increase in gel strength [45,46]. However, gel strength began to decrease when m-γ-PGA concentration exceeded 0.12%. It may be because m-γ-PGA concentration was too high, leading to excessive crossing linkages among protein molecules and preventing a stable gel network structure from forming, hence the deterioration of the gel strength [47,48].

### 3.5. Effect of m-γ-PGA Content on Gel Texture Properties

TPA simulates the chewing pattern of a human mouth using an analyzer to reflect the texture characteristics of the measured sample [36]. As m-γ-PGA content increased, all TPA indexes of minced meat paste gel tended to increase initially and then decrease (Table 2). The texture of the gel improved to different degrees after adding a certain percentage of m-γ-PGA, and the springiness, gumminess, chewiness, hardness, and cohesiveness of the gel significantly increased when m-γ-PGA content increased to 0.12% and then decreased. This is in correlation with the results obtained from the gel strength experiments (Figure 3). The explanation for this trend is that the presence of m-γ-PGA causes an augmentation of various negative charges that protein molecules carry on their surfaces, which results in an increase in water-molecule–protein interactions under electrostatic repulsion, thereby causing an increase in the solubility of myofibrillar proteins [49,50]. This provided more opportunities for protein–protein interactions during the subsequent heating process, contributing to the formation of protein gel and improving the quality of the meat gel [51]. In addition, a more stable and denser structure of the gel network may also be associated with cross-linking between myofibrillar proteins and m-γ-PGA [4,28]. Nevertheless, significantly high m-γ-PGA concentration can cause an overabundance of cross-linking between m-γ-PGA-protein and protein–protein. This could disrupt the ordering of the gel network, compromise its integrity, and ultimately affect the texture of the gel. These findings are consistent with those of Yu et al. [41] and Wang et al. [48].

### 3.6. Cooking Loss and WHC

Cooking loss and WHC reflect the ability of meat proteins to bind water and indirectly indicate the uniformity and stability of the meat gel structure, which are essential for assessing minced meat paste products’ quality [52]. In Figure 4, a noticeable decrease in cooking loss of meat gels was observed as the m-γ-PGA content increased from 0 to 0.12%. There was an inverse relationship between the WHC and cooking loss. At 0.12% m-γ-PGA addition, cooking loss and WHC reached minimum and maximum values of 15.80% and 79.91%, respectively, suggesting that an appropriate level of m-γ-PGA could dramatically reduce water loss in meat gels. This is because the abundance of unbound carboxyl groups on the side chains of m-γ-PGA have an extremely strong water-absorbing capacity [26]. Moreover, m-γ-PGA increases the system’s negative charge and enhances the electrostatic repulsion [53], causing molecular gaps between protein molecules to increase, thus retaining more water. On the other hand, m-γ-PGA may interact with meat proteins, forming a relatively dense and stable network structure, thus reducing water loss [44]. However, when m-γ-PGA was added at an amount greater than 0.12%, WHC began to decrease, and cooking losses began to increase. This may be because a high concentration of γ-polyglutamic acid affects the force between water and protein molecules, thereby weakening the capacity of proteins to bind with water. The WHC of the gels in this study displayed a similar trend to the hardness of meat gels.

### 3.7. Distribution of Water in Gel

LF-NMR can quickly and nondestructively determine the T_2_ relaxation time of samples, thus reflecting the state of water within meat gels [54]. All samples displayed four characteristic peaks: T_2b1_ (~0–1 ms); T_2b2_ (~1–10 ms); T_21_ (~10–150 ms); and T_22_ (~150–1000 ms), which were related to the three types of water molecules (Figure 5a). They represented strongly and weakly bound water (tightly bound to macromolecules by molecular forces), immobile water (retained within the protein gel network structure), and free water (present outside the gel network) in correlation with the meat gel relaxation times observed by Wu et al. [33]. T_2_ relaxation time reflects the fluidity of water. The shorter the time, the less mobile the water, and the tighter the bond between the water and the substrate. As shown in Figure 5b, T_2b1_ relaxation times for all the samples did not exhibit statistically significant differences, indicating that the strongly bound water was in a relatively stable state and did not migrate outward. With the m-γ-PGA content rising, T_2b2_ relaxation time decreased and then increased. This tendency was similar to the findings presented by Xu et al. [55]. The decrease in T_2b2_ indicated weaker water mobility and a lower migration rate. The reason may be that the formation of aggregates by m-γ-PGA with proteins provides more water molecule binding sites, which improves the capacity of proteins to combine with water [33]. As the dosage of m-γ-PGA was raised to 0.16%, T_2b2_ began to increase. This may be because high m-γ-PGA concentrations introduced side chains with a large number of highly reactive free carboxyl groups, which competed excessively with the proteins’ hydrophilic groups, and the hydrophobic effect prompted the rearrangement of water molecules around the proteins, with a reduced binding force between the proteins and water molecules [34]. When m-γ-PGA content increased to 0.12%, T_21_ and T_22_ relaxation times became the lowest. This may be because a denser gel network was formed, limiting water migration and reducing water loss. And when the content of m-γ-PGA addition continued to increase, water freedom became larger, and the gel structure was progressively disordered. This can explain the increase in T_22_ relaxation time [56].

In Figure 5c, P_2b_, P_21_, and P_22_ indicate the proportions of the total area occupied by the peak areas of T_2b_, T_21_, and T_22_, respectively. The immobile water area (P_21_) accounted for more than 90% of all the samples, demonstrating that the gel samples predominantly contained less mobile water, which is consistent with previous studies. When m-γ-PGA was added between 0 and 0.12%, P_21_ and P_22_ showed increasing and decreasing trends, respectively. This suggested that a fraction of the free water underwent conversion into immobile water within the meat gels. The significant water absorption of m-γ-PGA and gel’s stable, dense network structure could reduce water loss and improve water retention. However, as the m-γ-PGA content exceeded 0.12%, the structure of the gel network gradually became less orderly and stable, and the water mobility increased, so P_21_ began to decrease, and P_22_ began to increase. These results also provide an explanation for the cooking loss and WHC results in this study.

### 3.8. Color Evaluation

For consumers, color is an intuitive criterion for choosing meat products. The color of meat products is primarily affected by the content and state of myoglobin and hemoglobin in the muscle, while the moisture content, auxiliary materials, and other factors also change the color of meat products [4]. Table 3 demonstrates the colors of meat gels with different m-γ-PGA contents. The a* and b* values did not change much when the m-γ-PGA level changed from 0 to 0.2%, while with increasing m-γ-PGA content, the gels’ L* and W values exhibited a tendency to increase and then decrease. With the m-γ-PGA addition of 0.12%, the L* and W values reached a maximum of 64.81 ± 0.15 and 62.54 ± 0.20, respectively. This is probably because m-γ-PGA intensified the gel network density and stability, allowing the gel network to trap more water molecules. The light scattering of the gel was enhanced by higher water content, resulting in increased L* and W values. These findings were consistent with those of previous WHC results [4,57]. When the m-γ-PGA concentration continued to increase, the gels’ L* and W values began to decrease and reached a minimum with the 0.2% addition. On the one hand, it might be that the increase in m-γ-PGA concentration accelerated the Maillard reaction and the generation of colored substances. However, it was possible that the disordered gel network structure led to diminished light scattering due to water loss. This resulted in decreases in the L* and W values of the gel.

### 3.9. Microstructural Properties

Figure 6 illustrates alterations in the microstructures of minced beef paste gels following the incorporation of varying m-γ-PGA concentrations, where the black areas in the figure indicate the pores formed after dehydration and degreasing of the gels. The microstructure of the gel exhibited a strong correlation with both the WHC and texture of ground beef gel products [36]. The control without added m-γ-PGA (Figure 6a) had an irregular and discontinuous network structure, and the pores were not dense and homogeneous. This may have led to moisture loss and deterioration of the textural structure [57]. After adding m-γ-PGA, the gel network structure exhibited a progressive increase in density and uniformity, while the pores gradually became more homogeneous and refined. When m-γ-PGA concentration reached 0.12% (Figure 6c), the gel samples showed a highly homogeneous and compact network structure, which might be attributed to the electrostatic interactions facilitated by m-γ-PGA, thereby enhancing the cross-linking of the proteins [44]. When the m-γ-PGA concentration continued to increase, the pores began to become larger, and the network structure became rough, which might be because high m-γ-PGA concentration had too many anionic groups, resulting in excessive electrostatic repulsion destroying the network structure, and another explanation could be that high m-γ-PGA concentration prompted excessive aggregation and enhanced protein molecules cross-linking, disintegrating the network structure [47]. The microstructural findings were aligned with the water retention and gel strength characteristics.

### 3.10. Principal Component Analysis (PCA) and Correlation Analysis

Gel water-holding capacity (cooking loss, water-holding capacity), gel water distribution (P_2b1_, P_2b2_, P_21_, P_22_, T_2b1_, T_2b2_, T_21_, T_22_), gel textural properties (cohesiveness, hardness, chewiness, springiness, gumminess, gel strength), and gel color (W, L*, a*, b*) were investigated using PCA, as shown in Figure 7a. The PCA loading plot can reflect the magnitude and direction of the influence of the metrics on the principal components in the plot [58]. As shown in Figure 7a, PC1 accounted for 61.9% of the total variance, while PC2 accounted for 12.3%, and the overall contribution of the two amounted to 74.2%, indicating that there was a close relationship between the above indicators. Gel textural properties, water-holding capacity, and P_21_ positively affected PC1. Cooking loss, P_22_, T_2b2_, T_21_, T_22_, and b* negatively affected PC1. PC2 was positively affected by T_2b1_, L*, and W and negatively affected by P_2b1_, P_2b2_, and a*. Overall, there was a positive correlation between water-holding capacity, gel textural properties, and P_21_ and a negative correlation with P_22_, T_2b2_, T_21_, and T_22_. It indicates that both the intact and homogeneous gel structure and more content of water that is not easy to flow increase the water-holding capacity of the gel samples. The correlation between the metrics was further investigated using Pearson correlation analysis. As shown in Figure 7b, the correlation between WHC and P_2b2_, P_21_, and gel strength was significantly positive, whereas a significant negative correlation was observed between WHC and P_22_, T_2b2_, T_22_, and cooking loss. This suggests that the incorporation of m-γ-PGA caused an improvement in the meat gels’ texture and the ability to retain water, possibly because m-γ-PGA promoted a more orderly and dense gel formation, resulting in a more significant ability to limit water migration, which was in correlation with the findings of the PCA (Figure 7a).

## 4. Conclusions

In this study, medium molecular weight γ-PGA has good moisturizing properties, and a small amount of m-γ-PGA can significantly affect (*p* < 0.05) gel quality and rheology of minced beef paste. Compared with the blank group, 0.12% m-γ-PGA significantly improved water-holding capacity, texture hardness, elasticity, cohesion, and chewability. These changes are caused by certain properties of m-γ-PGA that make the gel network more stable and denser, thereby increasing gel strength and reducing water migration. Overall, our results suggest that m-γ-PGA is a safe hydrophilic material and possesses extensive potential for enhancing the quality of minced meat products.

## Figures and Tables

**Figure 1 foods-13-00510-f001:**
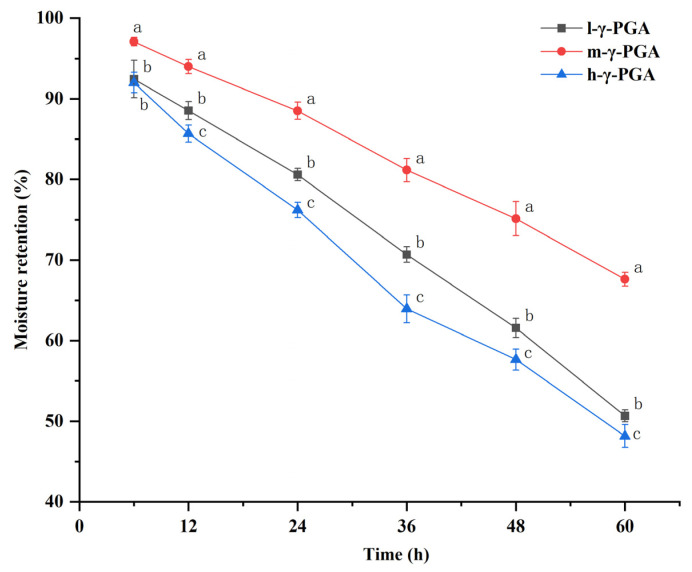
Moisturizing properties of γ-PGA degradation products. Different superscript letters (a–c) indicate significance (at α = 0.05 level); the same below.

**Figure 2 foods-13-00510-f002:**
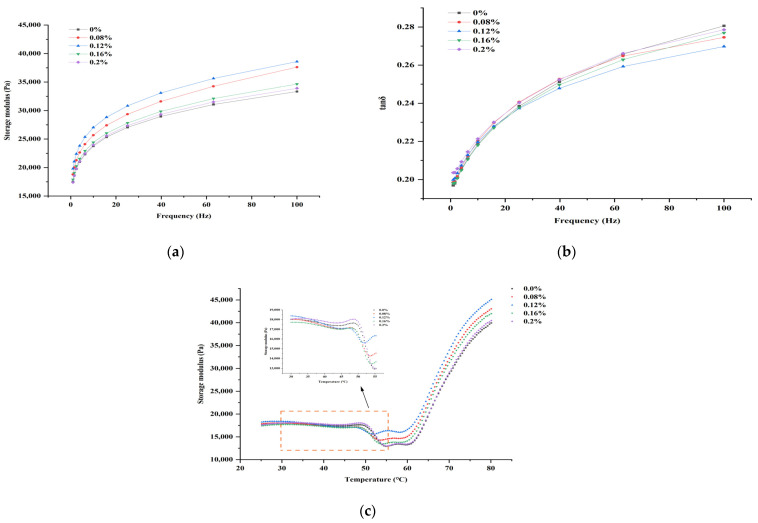
Effect of different additions of m-γ-PGA (0.00, 0.08, 0.12, 0.16, and 0.20%) on the storage modulus (**a**) and tan δ (**b**) at different frequencies and storage modulus at different temperatures (**c**) of minced beef meat paste.

**Figure 3 foods-13-00510-f003:**
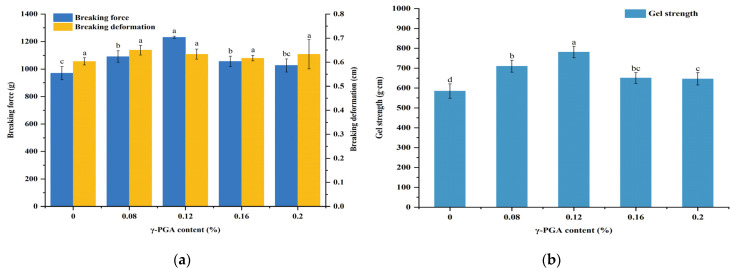
Effect of different additions of m-γ-PGA (0.00, 0.08, 0.12, 0.16, and 0.20%) on breaking force and deformation (**a**) and gel strength (**b**) of gels prepared from raw minced beef meat paste. Different superscript letters (a–d) indicate significance (at α = 0.05 level).

**Figure 4 foods-13-00510-f004:**
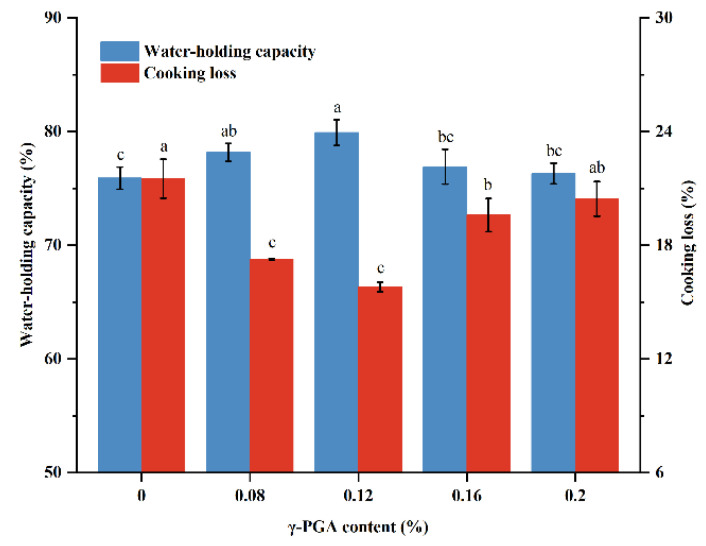
Effect of different additions of m-γ-PGA (0.00, 0.08, 0.12, 0.16, and 0.20%) on water-holding capacity and cooking loss of minced beef meat paste. Different superscript letters (a–c) indicate significance (at α = 0.05 level).

**Figure 5 foods-13-00510-f005:**
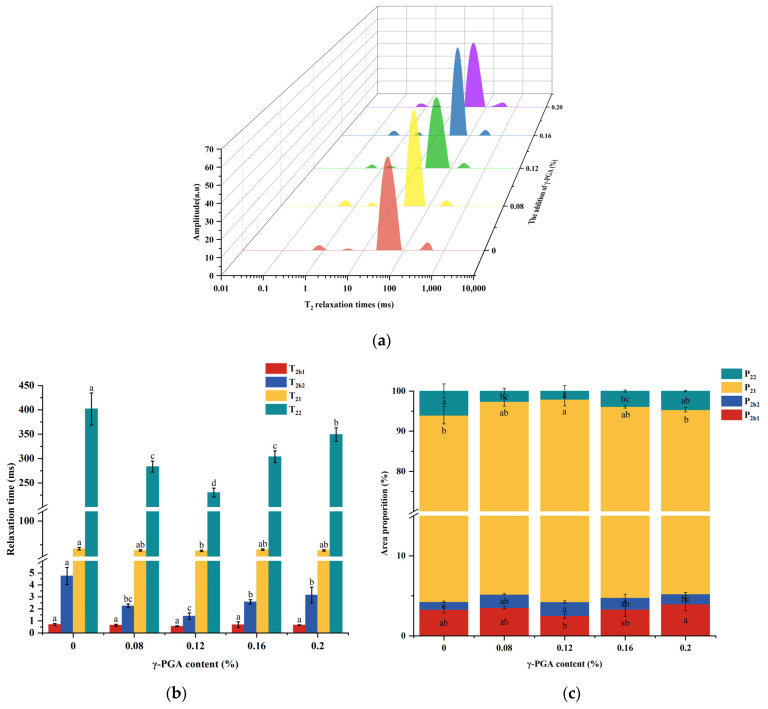
T_2_ relaxation time curves (**a**), T_2_ relaxation times (**b**), and T_2_ percentage content (**c**) in minced beef meat paste gels with different levels of m-γ-PGA added. The red, yellow, green, blue, and purple curves in Figure 5a represent the T_2_ relaxation time curves of meat gels with m-γ-PGA content of 0, 0.08, 0.12, 0.16, and 0.20%, respectively. Different superscript letters (a–d) in (**b**,**c**) indicate significance (at α = 0.05 level).

**Figure 6 foods-13-00510-f006:**
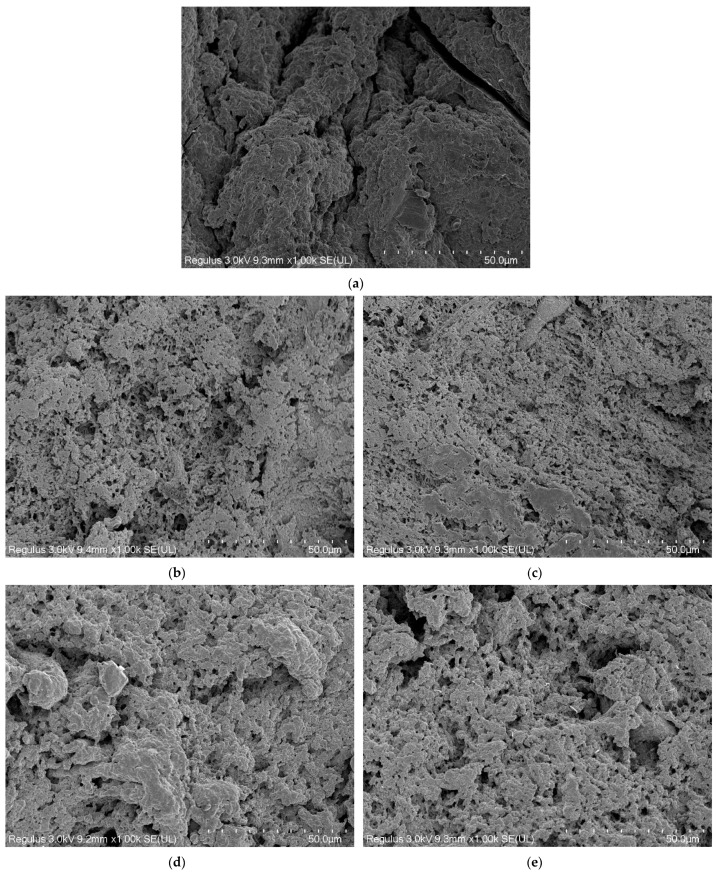
Microstructure of beef meat gels at m-γ-PGA concentrations of 0.00, 0.08, 0.12, 0.16, and 0.20% (**a**–**e**).

**Figure 7 foods-13-00510-f007:**
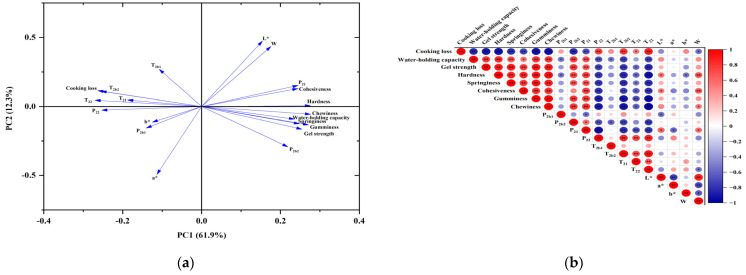
Principal component analysis (**a**) and Pearson’s correlation analysis (**b**). The arrows in (**a**) correspond to the loadings of the correlation indicators. The circle size and color gradient in (**b**) indicate the correlation coefficients (r-value), and the * and ** symbols in (**b**) represent significant correlation at the significance levels of 0.05 and 0.01, respectively.

**Table 1 foods-13-00510-t001:** Molecular weight distribution of γ-PGA.

Samples	Mn (Da)	Mw (Da)	PDI
l-γ-PGA	155,484	340,963	2.19
m-γ-PGA	565,275	731,502	1.29
h-γ-PGA	1,240,501	1,303,672	1.05

**Table 2 foods-13-00510-t002:** Textural parameters of minced meat gels with different additions of m-γ-PGA.

m-γ-PGA Content (%)	Hardness (g)	Springiness	Cohesiveness	Gumminess (g)	Chewiness (g)
0	6008.31 ± 105.74 ^e^	0.88 ± 0.01 ^d^	0.68 ± 0.01 ^c^	4093.49 ± 62.15 ^d^	3780.67 ± 71.50 ^e^
0.08	7011.90 ± 60.65 ^b^	0.91 ± 0.01 ^ab^	0.71 ± 0.02 ^ab^	5184.90 ± 70.37 ^b^	4683.89 ± 82.90 ^b^
0.12	7995.10 ± 84.85 ^a^	0.92 ± 0.00 ^a^	0.73 ± 0.01 ^a^	5437.57 ± 71.53 ^a^	4989.21 ± 43.46 ^a^
0.16	6564.22 ± 10.26 ^c^	0.90 ± 0.00 ^c^	0.69 ± 0.02 ^bc^	4727.90 ± 58.57 ^c^	4413.54 ± 58.76 ^c^
0.20	6307.70 ± 91.44 ^d^	0.90 ± 0.01 ^bc^	0.69 ± 0.01 ^bc^	4673.07 ± 83.47 ^c^	4084.28 ± 86.60 ^d^

Different letters (a–e) within the same column represent the significance of the difference (at α = 0.05 level).

**Table 3 foods-13-00510-t003:** Color of minced meat gels at different additions of m-γ-PGA.

γ-PGA Content (%)	L*	a*	b*	W
0	64.45 ± 0.34 ^ab^	3.00 ± 0.34 ^a^	13.55 ± 0.36 ^ab^	61.83 ± 0.45 ^b^
0.08	64.40 ± 0.32 ^ab^	3.08 ± 0.14 ^a^	13.70 ± 0.28 ^a^	61.73 ± 0.22 ^b^
0.12	64.81 ± 0.15 ^a^	2.87 ± 0.05 ^a^	12.50 ± 0.37 ^b^	62.54 ± 0.20 ^a^
0.16	64.13 ± 0.33 ^b^	3.05 ± 0.30 ^a^	14.23 ± 0.31 ^a^	61.29 ± 0.41 ^bc^
0.20	63.43 ± 0.26 ^c^	3.26 ± 0.12 ^a^	13.32 ± 1.06 ^ab^	60.94 ± 0.40 ^c^

Different letters (a–c) within the same column represent the significance of the difference (at α = 0.05 level).

## Data Availability

The data presented in this study are available on request from the corresponding author. The data are not publicly available due to the data are not publicly available due to privacy restrictions.

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
