# Peer review of "Minced Beef Meat Paste Characteristics: Gel Properties, Water Distribution, and Microstructures Regulated by Medium Molecular Mass of γ-Poly-Glutamic Acid"

_foods, 2024, doi:10.3390/foods13040510_

Round 1

Reviewer 1 Report

Comments and Suggestions for Authors

The study focuses on the influences of various m-γ-PGA (0.08–0.20%, w/w) concentrations on the properties of beef meat batters in terms of rheological properties, texture, moisture distribution, and microstructures. Overall, the work is well-written. However, some of the following points need to be considered and addressed.

1.      Line 56-67: these statements are not related to your study as the function and effects of γ-PGA on Beef meat are different than yogurt and plants. Please rephrase your statement and cite the relevant studies related to beef.

2.      a lack of baseline information on why the authors used γ-PGA on the properties of beef meat batters in terms of rheological properties, texture, moisture distribution, and microstructures. Please summarize the work's goals and provide relevant background information while avoiding a detailed literature review in the introduction.

3.      Why did you select m-γ-PGA at 0.08–0.20%, w/w concentrations on the properties of beef meat batters?

4.      The statements in Lines 328-331 required citations to support the explanation. 

Reviewer 2 Report

Comments and Suggestions for Authors

Dear authors,

My comments are attached.

Comments on the Quality of English Language

Minor editing is required.
